# Knowledge of postpartum complications and associated factors among women who gave birth in the last 12 months in Arba Minch Town, Southern Ethiopia, 2019: A community-based cross-sectional study

**Godana Yaya Tessema**[1]*, **Gistane Ayele**[2☯], **Kassahun Fikadu Tessema**[1☯], **Gebresilasea Gendisha Ukke**[1☯], **Wanzahun Godana Boynito**[2☯]

**1** Department of Midwifery, College of Medicine and Health Sciences, Arba Minch University, Arba Minch, Ethiopia, **2** School of Public Health, College of Medicine and Health Sciences, Arba Minch University, Arba Minch, Ethiopia

☯ These authors contributed equally to this work.
* godanayaya8@gmail.com, godana.yaya@amu.edu.et

**Data Availability Statement:** All relevant data are within the paper and its Supporting Information files. But if there is additional data required, I can

## Abstract

### Introduction

The knowledge of women about obstetric complications can helps them to seek health care earlier before obstetric complications arise. Most maternal deaths occur due to the poor health care seeking behavior after childbirth, but little is done on maternal knowledge of postpartum complications. Therefore this study aimed to assess knowledge of postpartum complications and associated factors among women who gave birth in the last 12 months in Arba Minch Town, Sothern Ethiopia.

### Methods

A community-based cross-sectional study was conducted on 418 women from December 01 to 15, 2019. A multi-stage sampling method was applied to reach study units. A semi-structured questionnaire was used to collect the data using face-to-face interviews. Bivariable and multivariable logistic regression was applied to examine the relationship between dependent and independent variables. Statistical significance was declared at a P-value ≤ 0.05 with the corresponding 95% confidence level.

### Results

Knowledge of women on postpartum complications was 23.9%. Secondary and above educational level (AOR = 3.82, 95% CI: [1.70, 8.65]), Grand multiparity (AOR = 2.31, 95% CI: [1.13, 4.71]), having four and above ANC visit (AOR = 2.04, 95% CI: [1.10, 3.81]) and self-decision making power to seek care (AOR = 3.68, 95% CI: [2.21, 6.11]) were statistically significant factors.

able to provide details of any required data elsewhere, without limitations. ORCID ID-0000-0002-4673-6975.

**Funding:** Author received specific funding from Arba Minch University for this work.

**Competing interests:** The authors have declared that no competing interests exist.

## Conclusion and recommendation

Mothers' knowledge of postpartum complications was low in this study area. Improving women's educational level, decision-making power to seek health care, and counseling during ANC follow-up may be useful approaches to increase their knowledge of postpartum complications.

## Introduction

The postpartum period is a time that starts one hour after delivery and extends up to 42 complete days postpartum. It is a critical phase in the lives of mothers and newborns. During this time, a wide range of postpartum complications have been reported, including excessive or prolonged postpartum bleeding, breastfeeding problems, urinary incontinence, constipation, depression, psychoses, post-traumatic stress disorder, anxiety, fatigue, constipation, and sleep disorders [1, 2].

Even though major changes and complications that threaten the lives of mothers and newborns occur during this period, there is a lack of appropriate care during this period, which results in maternal morbidity and mortality. The majority of maternal deaths occur within the first month of childbirth, with nearly half occurring within the first 24 hours and 66% occurring within the first week [3].

In 2015, around 303, 000 women died due to pregnancy and childbirth-related complications in the world. Almost all (99%) maternal deaths occur in developing countries, of which 66% of maternal deaths occur in Sub-Saharan African countries, which are estimated to have 546 maternal deaths per 100,000 live births. Most of these deaths could have been prevented if mothers had sought health care before complications worsened [4, 5].

The new transformative agenda set by the United Nations (UN) Sustainable Development Goal (SDG) is planned to reduce the global maternal mortality ratio (MMR) to less than 70 per 100,000 live births by 2030 [6].

Among pregnancy and childbirth-related complications, hemorrhage was the leading direct cause of maternal death worldwide, representing 27.1% of maternal deaths. Postpartum hemorrhage alone accounted for 19.7% of total maternal deaths due to hemorrhage. Hypertension was the second most common direct cause worldwide. Maternal mortality due to sepsis was 10.7% and other deaths were due to other direct causes [7].

According to the Ethiopian Demographic Health Survey (EDHS), the estimated pregnancy-related mortality ratio is 412 deaths per 100,000 live births. In Ethiopia, for every 1,000 live births, approximately four women die during pregnancy, childbirth, or within 2 months after childbirth. Although Ethiopia is one of the countries that makes progress toward achieving Mellenium Development Goal (MDG) 5, still Maternal Mortality Ratio (MMR) in the country is high [5, 8].

The postpartum period is associated with significant biosocial changes in women's lives. When these changes occur, they can be associated with problems such as vaginal bleeding foul-smelling vaginal discharge, constipation, tenderness and pain in the perineum, breast problems, burning during urination, wound breakdown, self-consciousness, and loneliness. If these problems progress, they can lead to complications that can negatively affect women's health and daily life after the postnatal period. Women's knowledge of these associated problems is deemed necessary to lower postpartum complications [9].

Many factors influence the outcome of pregnancy starting with the onset of any obstetric complications. Delayed treatment has the greatest negative impact on the outcome, which is

due to a variety of factors, including a lack of information and adequate knowledge about complications. According to one study, 80 percent of maternal deaths in Ethiopia occurred at home [10].

"Type 1 Delay " is the leading cause of maternal mortality in Ethiopia, and the main cause of "type 1delay" is a lack of knowledge about pregnancy-related issues. Women's knowledge is important for seeking medical care before complications arise and has a crucial influence on the reduction of maternal morbidity and mortality [11].

Despite several types of research being conducted on maternal knowledge of obstetric complications and associated factors, little is done on three separated phases of obstetrics, i.e., pregnancy, childbirth, and postpartum periods. In this phase, maternal health care as well as their knowledge of complications specific to this specific period is different. Therefore, this study was focused on women's knowledge of postpartum complications and associated factors.

## Methods & material

### Study area and period

The study was conducted from 1$^{st}$ December to 15$^{th}$ December 2019 in Arba Minch town. Arba Minch is the administrative town of Gamo Zone, South Nation Nationality People Region (SNNPR), and Ethiopia. It is found at an elevation of 1285 meters above sea level and is located 505 km away from Addis Ababa, the capital city of Ethiopia. People with different ethnic groups and religions reside in the town. The town is divided into four sub-cities, namely, Abaya, Sikela, Nech-Sar & Shecha. The town consists of 11 kebeles (the smallest administrative unit) with a total population of 112, 724. There are 26, 265 reproductive age group (15–49) women who reside in the town out of which 4428 were pregnant. The total number of mothers who gave birth at the facility preceding the study period was 3992. There is one governmental General Hospital, two Health Centers, 17 primary care clinics, and 14 medium-sized private clinics in the town [12].

### Study design

A community-based cross-sectional study.

### Population

All women who gave birth in the last 12 months before the start of the study and lived in Arba Minch town for six months were chosen. Individual mothers were systematically selected and interviewed in four randomly selected kebeles, with a total of 1475 mothers eligible for the study sample allocation. Mothers who were critically ill and unable to participate in the interviews were excluded from the study. As a result, the study included those mothers who had given birth within the previous year before the study period.

### Sample size determination and sampling procedures

**Sample size determination.** The sample size was calculated by using a single population proportion formula with the assumption of, a 95% confidence level, 5% margin of error, 5% non-response rate, 1.5 design effect, and expected population proportion of mothers who were

knowledgeable about postpartum complications was 22.1% [13].

$$n = \frac{\left(z_{\frac{\alpha}{2}}\right)^2 \cdot p(1-p)}{d^2}$$

Where,

   n = sample size

   Z = standard normal distribution value at 95% confidence level of $\frac{\alpha}{2}$ = 1.96

   P = women's knowledge of postpartum complications = 22.1% [13].

   d = margin of error = 5%

$$n = \frac{(1.96)^2 \times 22.1(1-0.221)}{0.05^2}$$

**n** = **265**

Based on the assumptions the final calculated sample was: **n = 418**

**Sampling procedures.** Arba Minch town was purposively selected as a study area. The study participants were chosen using a multistage sampling process. There were 11 kebeles in the town, so kebeles were divided into 11 clusters. Using a simple random sampling procedure, four kebeles (Bere, Ediget-Ber, Wuha-Minch, and Dilfana) were chosen, and mothers who gave birth in the previous 12 months in each kebele were identified by contacting kebele health extension workers (HEW) for the updated register. The overall number of mothers who gave birth in the four kebeles was N (estimated total number of deliveries in four kebeles) 1475 (Bere = 331, Ediget-ber = 471, Wuha-Minch = 371 & Dilfana = 302 ladies). Using a proportional allocation formula, the sample assigned to a specific kebele was considered. The calculated samples for Bere, Ediget Bere, Wuha Minch, and Dilfana were 94, 133, 105, and 86 mothers, respectively, based on the proportion to size allocation. The 'K'-value was calculated by dividing the number of mothers who gave birth in the previous 12 months in each selected kebele by the sample allocated to that kebele, yielding the following results: 3.52, 3.54, 3.53, and 3.51 for Bere, Ediget Ber, Wuha Minch, and Dilfana, respectively. By chance, the rounded k-value of all of the chosen kebeles was 3. The first mother was discovered by randomly selecting one household on the kebele's outskirts with the assistance of HEW and the leaders of specific kebeles. Then, for every three mothers, a systematic random sampling technique was used to obtain study participants, which was repeated until the sample was complete. If more than one mother was found in a single household using the lottery method, only one mother was chosen.

## Data collection tools and procedures

The data were collected by face-to-face interviews using a semi-structured questionnaire. The questionnaire was adapted from previously approved types of literature considering the study objectives [17, 18]. It consists of socio-demographic variables, obstetric factors, and postpartum complications knowledge measuring questions. The questionnaire was prepared in English and was translated to Amharic (working language of the study area) and then, back to English for consistency.

   Three female diploma midwives who are fluent in speaking the Amharic language were involved in data collection. Two males who have Bachelors of Science degrees in midwifery' health professionals were recruited as supervisors.

## Study variables

**Dependent variable.** Women's knowledge of postpartum complications.

**Independent variables.** *Socio-demographic variables*. Maternal age, marital status, occupational status, educational status, monthly income **&** access to media.

*Obstetric factors & health service use*. Parity, Antenatal care visit, information on the pregnancy-related problem, place of delivery, self-decision making autonomy to seek health care.

## Operational definition

**Knowledgeable.** Mothers who spontaneously mentioned three and above postpartum complications were declared as knowledgeable about postpartum complications [13, 14].

## Data quality management

The quality of the data was maintained before, during, and after data collection.

A semi-structured questionnaire was adapted from different types of previously published literature. Then pretesting of the questionnaire was carried out on 5% of the sample that was, on 21 mothers in Mirab Abaya town, Gamo Zone, Ethiopia, and any necessary amendments were done.

Intensive training of three days duration about the objective of the study, questionnaire, and ethical issues was given to data collectors and supervisors.

During the data collection period, questionnaires were checked for completeness and consistency immediately by data collectors.

After data collection, the principal investigator and supervisors rechecked the collected data for its completeness, and corrective measures were taken accordingly. Then, data was entered to epi info version 3.5.1 and 5% of the data set was double entered to check the accuracy of the entered data.

## Data processing and analysis

After data collection was complete, each questionnaire was coded and entered into Epi info version 3.5.1 and then, it was exported to SPSS Version 20 window compatible software for cleaning and analysis. By using simple frequency tables and cross-tabulation data were checked for completeness.

A descriptive statistical method such as frequency tables, graphs, and mean with standard deviation was used to present different characteristics of study participants.

Bivariable logistic regression was done and variables with P-value $\leq 0.25$ and all other variables that had an association in previously approved kinds of literature and assumed to have scientific relevance to the study were selected as candidate variables for multivariable logistic regression analysis to control the effect of confounders. A backward stepwise logistic regression method was used for the analysis. Model fitness was checked by Hosmer and Lemeshow's goodness of model fit test (P = 0.434).

Finally adjusted odds ratio with corresponding 95% CI and p-value $\leq 0.05$ was taken as a statistical association between the dependent and independent variables.

## Ethical consideration

Ethical approval for the study was obtained from Arba Minch University, College of Medicine and Health Sciences Institutional Research Review Board (IRB/132/12 dated December 27, 2019). A formal letter of permission to conduct the study was obtained from Gamo Zone Health Department, Arba Minch Town Health Office. The verbal consensus was made with

mothers after an explanation of the purpose of the study, and the part they took in the research by data collectors to assure their right to refuse or participate in the study. Mothers were also told that the data obtained will be kept confidential & had the right to withdraw from the study at any time during the interview.

## Results

### Socio-demographic characteristics of study participants

A total of 418 women who gave birth in the twelve months preceding the study period were recruited to participate and resulting in a 100% response rate. The respondents' average age was 28.35 (SD 5.84) with a minimum and maximum of 18 and 42 ages respectively. The majority of respondents 202(48.3%) and 181(43.3%) were Orthodox and Protestant, respectively, and 400(95.7%) were married. The majority of respondents, 267 (63.9%), were of Gamo ethnic group and 238 (56.9%) were housewives. In terms of level of education, 222 (53.1%) of respondents have completed Secondary School or higher. Among the total study participants, 367 (87.8%) had access to mass media, with television being the most commonly used medium 335 (80.1%) (Table 1).

**Obstetric characteristics of mothers and health service utilization.**   Among the 418 mothers polled, 188 (45%) were multiparous, and 100 (23.9%) were grand multiparous. 411 (98.3 percent) of the total study participants attended ANC follow-ups for their most recent pregnancy. 283 (67.7 percent) of mothers who had a history of ANC visits had attended four or more times. The majority of the attendees had visited government facilities, with only 6 (1.4 percent) having visited a private clinic. Participants in the study were also asked whether or not health care providers provided information on pregnancy-related issues during visits to the ANC clinic. 275 (65.8 percent) of respondents received information about pregnancy-related problems from a health care provider, while 136 (32.5 percent) did not. The vast majority of mothers, 392 (93.8%), gave birth in a health facility, while 26 (6.2%) gave birth at home. Women were also asked if they were the primary decision-makers in their families when it came to seeking medical attention for pregnancy-related issues. Only 127 (30.4 percent) of total respondents were self-decision makers, and 291 (69.6 percent) of women's decisions were accompanied by their husbands and other family members (Table 2).

**Knowledge of postpartum complications.**   Even though the majority of the study participants 260(62.2 percent) had heard about postpartum complications before the study, only 100 (23.9 percent) of 418 women had Knowledge of postpartum complications (knew three or more postpartum complications). The main sources of information mentioned by respondents were: health personnel during ANC follow up 180 (43.1 percent), HEW (health extension workers) discussions on maternal health 143 (34.2 percent), mass media 83 (19.9 percent), school 18 (4.3 percent), and had previous history of obstetric complication 9 percent (2.2 percent). Severe vaginal bleeding (240, or 57.4 percent) was the most frequently mentioned postpartum complication, followed by pregnancy-induced hypertension after childbirth (121, or 28.9 percent). Breast complications were mentioned by 75 (17.9%), blurred vision and weakness were mentioned by 73 (17.5%), foul-smelling vaginal discharge was mentioned by 71 (17%), convulsions were mentioned by 14 (3.3%), and postpartum psychoses and blues were mentioned the least by 4 (1%) of the respondents. The majority of respondents who had heard about postpartum complications expressed a desire to visit health care facilities if complications arose (Fig 1).

**Factors associated with women's knowledge of postpartum complications.**   After multi-variable logistic regression was done; some variables such as secondary and above educational

**Table 1. Sociodemographic characteristics of the respondents in Arba Minch town, Southern Ethiopia, December 2019 (n = 418).**

| Variables | | Frequency(n = 418) | Percent (%) |
|---|---|---|---|
| **Age** | 15–19 Years | 18 | 4.3 |
| | 20–24 years | 97 | 23.2 |
| | 25–29 years | 136 | 32.5 |
| | ≥ 30 years | 167 | 40 |
| **Religion** | Orthodox | 202 | 48.3 |
| | Protestant | 181 | 43.3 |
| | Muslim | 35 | 8.4 |
| **Marital status** | Married | 400 | 95.7 |
| | Others* | 18 | 4.3 |
| **Ethnicity** | Gamo | 267 | 63.9 |
| | Amhara | 35 | 8.4 |
| | Konso | 31 | 7.4 |
| | Derashe | 24 | 5.7 |
| | Gofa | 21 | 5 |
| | Zeyissie | 18 | 4.3 |
| | Others** | 22 | 5.3 |
| **Occupation** | Gov't employee | 74 | 17.7 |
| | Merchant | 80 | 19.1 |
| | Housewife | 238 | 56.9 |
| | Other** | 26 | 6.2 |
| **Income** | < 500 ETB | 278 | 66.5 |
| | 500–1000 ETB | 43 | 10.3 |
| | 1001–1500 ETB | 18 | 4.3 |
| | ≥ 1501 ETB | 79 | 18.9 |
| **Women's educational** | Not attend formal education | 96 | 23 |
| | Primary | 100 | 23.9 |
| | Secondary &above | 222 | 53.1 |
| **Have media** | Yes | 367 | 87.8 |
| **Medias used by mothers** | Television | 335 | 80.1 |
| | Radio | 148 | 35.4 |
| | Newspaper | 16 | 3.8 |

*include Widowed, Divorced **Gurage, Oromo, Wolaita, Silte, & Kore, **Student.

status, grand multipara, four and above the frequency of ANC visit and decision making power of mothers to seek health care utilization were significantly associated with maternal knowledge of postpartum complications.

The educational status of the mothers showed statistical association with knowledge of postpartum complications, (AOR = 3.82, 95% CI: [1.70–8.65]).

Grand multipara mothers also had an association with the dependent variable, (AOR = 2.31, 95% CI: [1.13–4.71]). Similarly, frequency of antenatal care was another significant predictor of women's knowledge of postpartum complications, (AOR = 2.04, 95% CI: [1.10–3.81]).

Another variable that was associated with maternal knowledge of postpartum complications was the decision-making power of mothers to seek health service utilization, (AOR = 3.68, 95% CI: [2.21, 6.11]) (Table 3).

**Table 2. Obstetric characteristics and health service utilization of the respondents in Arba Minch town, Southern Ethiopia, December 2019.**

| Variables | | Frequency(n = 418) | Percent(%) |
|---|---|---|---|
| **Parity** | Primiparous | 130 | 31.1 |
| | Multiparous | 188 | 45 |
| | Grand Multipara | 100 | 23.9 |
| **Have ANC visit** | Yes | 411 | 98.3 |
| | No | 7 | 1.7 |
| **Frequency of ANC visit** | ≥ 4 visit | 283 | 67.7 |
| | 1–3 visit | 128 | 30.6 |
| **Institution of ANC visit** | Hospital | 221 | 52.9 |
| | Health center | 192 | 45.9 |
| | Private clinic | 6 | 1.4 |
| **Informed on pregnancy-related problem** | Yes | 275 | 65.8 |
| | No | 136 | 32.5 |
| **Place of delivery** | Health institution | 392 | 93.8 |
| | Home | 26 | 6.2 |
| **Decide to seek health care** | Self | 127 | 30.4 |
| | Husband/Family | 291 | 69.6 |

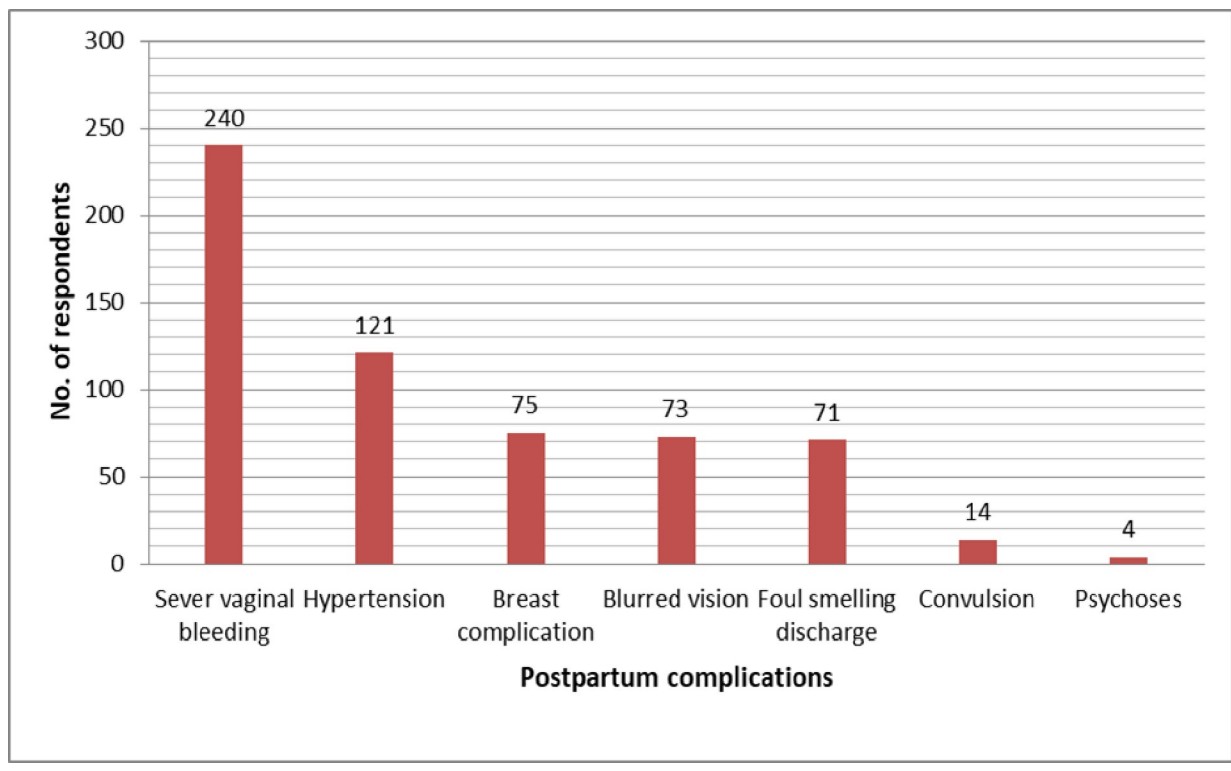

**Fig 1. Postpartum complications mentioned by women who gave birth in the last 12 months in Arba Minch town, Southern Ethiopia, December 2019.**

**Table 3.** Factors associated with mothers' knowledge of postpartum complications in Arba Minch town, Southern Ethiopia, 2019 (n = 418).

| Variables | Categories | Knowledge of postpartum complications | | COR (95% CI) | AOR(95%CI) |
|---|---|---|---|---|---|
| | | Yes | No | | |
| Age | 15–19 Years | 2 | 16 | 1 | *** |
| | 20–24 years | 16 | 81 | 1.58(0.33–7.56) | |
| | 25–29 years | 35 | 101 | 2.32(0.61–8.82) | |
| | ≥ 30 years | 47 | 120 | 3.13(0.69–14.16) | |
| Occupation | Gov't employee | 27 | 47 | 1.92(0.69–5.35) | *** |
| | Merchant | 20 | 60 | 1.11(0.39–3.15) | |
| | Housewife | 47 | 191 | 0.82(0.31–2.16) | |
| | Other**** | 6 | 20 | 1 | |
| Income | < 500 ETB | 56 | 222 | 1 | *** |
| | 500–1000 ETB | 10 | 33 | 1.20(0.56–2.58) | |
| | 1001–1500 ETB | 5 | 13 | 1.53(0.52–4.46) | |
| | ≥ 1501 ETB | 29 | 50 | 2.30(1.34–4.00) | |
| Educational status | Not attend formal education | 9 | 87 | 0.20(0.10–0.43) | **3.82(1.70–8.65)*** |
| | Attend primary school | 16 | 84 | 0.37(0.20–0.68) | 2.12(1.10–4.01) |
| | Secondary school and above | 75 | 147 | 1 | 1 |
| Access for media | Yes | 95 | 272 | 3.21(1.24–8.33) | *** |
| | No | 5 | 46 | 1 | |
| Parity | Primpara | 25 | 105 | 0.64(0.35–1.20) | **2.31(1.13–4.71)** |
| | Multipara | 48 | 140 | 0.93(0.54–1.61) | *** |
| | Grand multipara | 27 | 73 | 1 | **1** |
| ANC visit | Yes | 99 | 312 | 1.69(0.23–10.43) | *** |
| | No | 1 | 6 | 1 | |
| Frequency of ANC | ≥ 4 | 82 | 201 | 1 | **1** |
| | 1–3 | 17 | 111 | 0.38(0.21–0.67) | **2.04(1.10–3.81) *** |
| Place of birth | Health institution | 98 | 294 | 4.00(0.93–17.23) | *** |
| | Home | 2 | 24 | 1 | |
| Decision making power | 1 = self | 54 | 73 | 1 | **1** |
| | 2 = family and husband | 46 | 245 | 0.25(0.16–0.41) | **3.68(2.21–6.11)*** |

*Statistically significant variable at p-value < 0.01; ** significant variable at p-value <0.05; ***Variables which are not statistically significant in multivariable logistic regression ****Students.

## Discussion

This study reported that 23.9% of the study participants were knowledgeable on postpartum complications.

This study finding is much higher than a similar study conducted in Western Bengal of the Indian state, in 2011, in which the percentage of mothers who were knowledgeable on postpartum complications was 1.1% [15]. This difference could be due to variation in socio-demographical and time duration of the study.

The finding of a cross-sectional study carried out in rural area Gurage Zone, Ethiopia that reported 20.3% of mothers had mentioned two and above complications after childbirth, but in this study, 23.9% of mothers had mentioned three and above postpartum complications [16]. This study discrepancy might be due to the difference of study areas.

This study finding is slightly similar to cross-sectional studies conducted in two districts of the country; the Goba and Raya Kobo districts of Ethiopia that showed 22.1% & 26.4% of the mothers were knowledgeable on postpartum complications respectively [13, 14]. This consistence might be due to the similarity of the studies; in declaring mothers who had mentioned three and above complications after childbirth as knowledgeable.

The study finding is also less as compared with studies conducted in different parts of Ethiopia, such as Aleta Wondo Sidama (37.7%), Fango district Wolita (55.8%), & Debre Birhan Amhara (77.2%) [17–19]. This difference might be because mothers who could mention two and above complications were considered as knowledgeable in the previous study. But in the current study mothers were declared as knowledgeable if they mentioned three and above complications after childbirth.

This study revealed that severe vaginal bleeding was the most commonly known postpartum complication mentioned by study participants followed by hypertension, breast complication & foul-smelling discharge. This result is slightly in line with different research findings conducted in other countries as well as in Ethiopia [20–22]. The least mentioned complication was postpartum psychoses and blues which were never mentioned in other research findings. This might be due to maternal history of previous experience of the complication.

In this study; educational status, being grand multiparity, the number of ANC visits and the decision-making power of mothers to seek health care were predictors of maternal knowledge of postpartum complications.

The educational status of the mothers showed a statistical significant association with knowledge of postpartum complications. Women who attended secondary and above educational level were about 4 times (AOR = 3.82, 95% CI: [1.70, 8.65] more likely to be knowledgeable than mothers who didn't attend formal education. This was consistent with a research finding of Raya Kobo, Ethiopia [14].

Similarly, the frequency of antenatal care visit was another predictor of women's knowledge of postpartum complications. And women who had four and above ANC follow up were 2 times (AOR = 2.04, 95% CI: [1.10, 3.81]) more likely to be knowledgeable on postpartum complications than women who had three or fewer ANC visits. This study finding is similar to different studies conducted in Ethiopia [23, 24].

Being grand multiparity also has a relationship with maternal knowledge of postpartum complications. Grand multipara mothers were 2 times (AOR = 2.31, 95% CI: [1.13, 4.71]) more likely to have knowledge of complications of childbirth than primipara mothers. This is similar to the institutional-based cross-sectional study finding of Mechekel East Gojjam Zone, Ethiopia [25].

Another very important variable that was associated with maternal knowledge of postpartum complications, with a p-value of $< 0.001$ was the decision-making power of mothers to seek health service utilization. Mothers who decide health service utilization in their household had 4 times (AOR = 3.68, 95% CI: [2.21, 6.11]) a likelihood to be knowledgeable than mothers who depend on the decision of other family members.

This is in line with study findings conducted in Arba Minch town, Ethiopia; that studied knowledge of obstetric danger signs and associated factors [24].

## Limitation & strength of the study

Due to the nature of the cross-sectional study design, this study couldn't establish cause and effect relationships. This is the main limitation of the study.

As a strength, the study was conducted at the community level, which increases the generalizability to the entire urban area.

## Conclusion and recommendations

### Conclusion

According to this result mother's knowledge of postpartum complications is low. The most known complication after childbirth was severe vaginal bleeding followed by hypertension. Educational status, being grand multipara, the number of ANC visits and the decision-making power of mothers to seek health care were statistically associated variables that will predict maternal knowledge of postpartum complications.

### Recommendations

Improving women's educational status and decision-making power to seek health care before complications happened is needed. Increasing information delivery system about specific post-partum complications during antenatal follow-up is the recommended intervention. Based on the study findings the following concerned bodies should be recommended to improve and strengthen women's knowledge of postpartum complications.

- Health personnel and health extension workers should strengthen awareness creation activities by conducting maternal discussion forums at the community level.

- The Ministry of health should continue to encourage pregnant mothers to attend four and above antenatal visits.

- Health extension workers and community leaders should mobilize the society about the advantage of women's self decision-making power to seek health care at the household level.

- The Ministery of education Should continue to encourage women's education.

- Further research supported with qualitative data will be needed.

## Supporting information

**S1 File. English & Amharic version questionnaire.**
(DOCX)

## Acknowledgments

We are thankful to all the participants who invested their time and perspectives with us.

## Author Contributions

**Conceptualization:** Godana Yaya Tessema.

**Data curation:** Godana Yaya Tessema.

**Formal analysis:** Godana Yaya Tessema.

**Investigation:** Godana Yaya Tessema.

**Methodology:** Godana Yaya Tessema.

**Project administration:** Godana Yaya Tessema.

**Software:** Godana Yaya Tessema, Wanzahun Godana Boynito.

**Supervision:** Godana Yaya Tessema.

**Validation:** Wanzahun Godana Boynito.

**Writing – original draft:** Godana Yaya Tessema.

**Writing – review & editing:** Gistane Ayele, Kassahun Fikadu Tessema, Gebresilasea Gendisha Ukke, Wanzahun Godana Boynito.

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

27375920

