## [Decision Letter · Decision Letter 0]

24 Jan 2022

PONE-D-21-29865Knowledge of postpartum complications and associated factors among women gave birth in the last 12 months in Arba Minch Town, Southern Ethiopia, 2019: Community Based Cross-Sectional StudyPLOS ONE

Dear Dr. Yaya,

Thank you for submitting your manuscript to PLOS ONE. After careful consideration, we feel that it has merit but does not fully meet PLOS ONE’s publication criteria as it currently stands. Therefore, we invite you to submit a revised version of the manuscript that addresses the points raised during the review process.

Please submit your revised manuscript by  Mar 10 2022 11:59PM. If you will need more time than this to complete your revisions, please reply to this message or contact the journal office at plosone@plos.org. Please include the following items when submitting your revised manuscript:A rebuttal letter that responds to each point raised by the academic editor and reviewer(s). You should upload this letter as a separate file labeled 'Response to Reviewers'.A marked-up copy of your manuscript that highlights changes made to the original version. You should upload this as a separate file labeled 'Revised Manuscript with Track Changes'.An unmarked version of your revised paper without tracked changes. You should upload this as a separate file labeled 'Manuscript'.

We look forward to receiving your revised manuscript.

Kind regards,

Godwin Otuodichinma Akaba, MBBS,MSc,MPH,FWACS

Academic Editor

PLOS ONE

Journal Requirements:

2. Please amend your current ethics statement and your Methods to clarify whether written or verbal consent was obtained. Thank you.

Whilst you may use any professional scientific editing service of your choice, PLOS has partnered with both American Journal Experts (AJE) and Editage to provide discounted services to PLOS authors. Both organizations have experience helping authors meet PLOS guidelines and can provide language editing, translation, manuscript formatting, and figure formatting to ensure your manuscript meets our submission guidelines. To take advantage of our partnership with AJE, visit the AJE website (http://aje.com/go/plos) for a 15% discount off AJE services. To take advantage of our partnership with Editage, visit the Editage website (www.editage.com) and enter referral code PLOSEDIT for a 15% discount off Editage services.  If the PLOS editorial team finds any language issues in text that either AJE or Editage has edited, the service provider will re-edit the text for free.

We would like to acknowledge Arba Minch University College of Medicine and Health Sciences in providing with necessary financial & material and making the work successful.   

6. Please remove your figures from within your manuscript file, leaving only the individual TIFF/EPS image files, uploaded separately.  These will be automatically included in the reviewers’ PDF.

7. Please ensure that you refer to Figure 1 in your text as, if accepted, production will need this reference to link the reader to the figure.

8. We note you have included a table to which you do not refer in the text of your manuscript. Please ensure that you refer to Table 3 in your text; if accepted, production will need this reference to link the reader to the Table.

Additional Editor Comments:

Abstract:

Title: Knowledge of postpartum complications and associated factors among women gave birth in the last 12 months in Arba Minch Town, Southern Ethiopia, 2019: Community Based Cross-Sectional Study

Comment: Paraphrase to : Knowledge of postpartum complications and associated factors among women who gave birth in the last 12 months in Arba Minch Town, Southern Ethiopia, 2019: Community Based Cross-Sectional Study

Conclusion and recommendation: Conclusion and recommendation: Mother’s knowledge of postpartum complications was low in this study area. Empowering women educational level, decision making power to seek health care and counseling during ANC follow up will needed.

Comment: Paraphrase to: Mother’s knowledge of postpartum complications was low in this study area. Improving women’s educational level, decision making power to seek health care and counseling during ANC follow up may be useful approaches to increasing their knowledge of postpartum complications.

Introduction

This section has not been properly written as focus has not been adequately placed on knowledge of postpartum complications and its contribution to maternal mortality in Ethiopia.

I suggest that authors should revise this section as follows:

-A standard definition of the postpartum period.

-A highlight of the contribution of deaths in this period to maternal mortality globally, regionally and in Ethiopia.

-Description of the possible associated factors to deaths during this period including the relevance of knowledge about possible complications that could arise particularly life-threatening conditions occurring withing the first 24 hours of delivery and thereafter.

-Mention and cite previous literatures on the subject in Ethiopia and their limitations which justifies the need for the current study in Ethiopia.

-A statement on the objective of the current study

METHODS & MATERIAL:

Item: Study area and Period

Comment: Please paraphrase to:

The study was conducted from December 1- 15/ 2019 in Arba Minch town. Arba Minch is the administrative town of Gamo Zone, South Nation Nationality People Region (SNNPR), and Ethiopia. It is found at an elevation of 1285 meters above sea level and located 505 km away from Addis Ababa, the capital city of Ethiopia. People with different ethnic groups and religions reside in the town. The town is divided into four sub cities, namely, Abaya, Sikela, Nech-Sar & Shecha. The town consists of 11 kebeles (the smallest administrative unit) with a total population of 112, 724. There are 26, 265 reproductive age group (15-49) women who reside in the town out of which 4428 were pregnant. The number of facility childbirth was 3992. There is one governmental general hospital, two health centers and 17 primary, 14 medium private clinics in the town [12]

Study Design

Population

Item: All women who gave birth in the last 1 year in Arba Minch town were taken as source population. From four randomly selected Kebeles, individual mothers were selected by systematical and interviewed.

Comment: Paraphrase to: All women who gave birth in the last 1 year preceding the date of commencement of the study and reside in Arba Minch town were taken as source population. From four randomly selected Kebeles, individual mothers were systematically selected and interviewed. Mothers who were severely ill and unable to respond to the interviews were excluded from the study.

Comment: Please clearly state the number of women who were eligible for this study in the area of study (women who delivered in the last one year). This will be important in determining the sampling frame and systematic selection of participants as mentioned above. in How many of these mothers who were severely ill were actually excluded from the study? Is it possible these women may have had even poorer knowledge and thus may have affected the final results?

Item: Eligibility criteria Inclusion criteria

Mothers who gave birth in the last 1 year Months preceding the study period.

Who live in Arba Minch town for at least six Months?

Item: Exclusion criteria � Mothers those are severely ill and unable to respond.

Comment: Please delete the subsequent statements on eligibility and exclusion criteria as they may be seen as repetitions.

Item: Sample size determination

Comment: Please include the formula for sample size calculation and reference.

Item: Sampling procedures

. Among the Kebeles four Kebeles (Bere, Ediget-Ber, Wuha-Minch & Dilfana) were selected by simple random sampling method as a starting sampling technique.

Comment: It is not clear what authors meant by a ”starting sampling technique”.

Item: Firstly the number of mothers who gave birth in last 12 months in each selected Kebeles was determined contacting kebele health extension workers (HEW) for update registry. Then proportion to size allocation to each selected Kebeles was done.

Comments: What was the total number of women who gave birth in all the Kebeles? What is the number for each of the 4 kebeles? These needs to be included to help the reader appreciate how the allocation and sampling were carried out.

Study variables

Item: Operational definitions

Knowledgeable: Mothers who spontaneously mentioned three and above postpartum complications were declared to have knowledge about postpartum complication.

Comment: It is not clear how knowledge about three or more postpartum complications qualifies to be adequate knowledge. Apart from the quoted study, what informed the decision to peg it at three and not five? For most purposes and standards, a universal scoring system should have been applied. For example, using the academic rating scales of assessing students passing or failing an examination etc? Also list the total number of complications presented to patient in the questionnaire?

Item: Data quality management

Pretesting of the questionnaire was carried out on 5% (21) of sample in Mirab Abaya town, Gamo Zone, Ethiopia and any necessary amendments were done.

Comment: Please clarify on the (21) above.

Data processing and analysis

Item: Bivariable logistic regression was done and variables with P- value ≤ 0.25 and all other variables that had association in previously approved literatures were selected as candidate variables for multivariable logistic regression analysis to control the effect of confounders.

Comment: Please explain the rationale for inclusion of variables with P- value ≤ 0.25 instead of P- value ≤ 0.05 with relevant references. Also clarify regarding the quality of the previous studies that had variables with associations that were included in the multiple regression analysis.

Item: Ethics consideration

Comment: In the Ethics statement form on page 3, authors declared that written informed consent was taken from participants, ‘The letter of ethical clearance was obtained from Institutional Review Board (IRB) of College of Medicine & Health Sciences, Arba Minch University, Ethiopia. Also written consent was taken from study participants during data collection”.

In the ethics consideration section of the manuscript, it is noted that the authors have declared differently that verbal informed consent was obtained from mothers.

It is not clear which of the two was done at this point. Authors should clearly highlight this important point in the revision.

RESULT

Item: A total of 418 women who gave birth in the last twelve months preceding the study period were designed to participate in the study, this gives 100% response rate.

Comment: From the above comment it appears that they were enlisted to participate in the study. Is 418 the total number of women who gave birth during the study period? Did all participate in the study? Please describe the patient’s recruitment and participation using a flow chart as figure 1.

Result

This section needs to be revised taking into consideration the need to correct grammatical errors and delete abbreviations like &.

Item: Knowledge on postpartum complications

Majority of the study participants 260(62.2%) had information about postpartum complications.

Comment: It is not clear the information about postpartum complications that is being referred to here. What is the authors definition of having information? Is it the same as knowledge? How was this variable ascertained?

Perhaps it is better to say: Majority of the study participants 260(62.2%) had heard about postpartum complications before the study.

Discussion

This section lacks in-depth analysis of study findings particularly as it relates to the objectives of the study. Authors have not considered a critical evaluation of the studies they have tried to compare their studies with to find out if there are similar patients characteristics, sample sizes, similar assumptions etc but have majorly attributed difference or similarities in findings to either location, sociodemographic etc. They have not identified possible specific factors for example in sociodemographic characteristics that are different or similar. Public health implication for the study area and Ethiopia should be highlighted.

Reviewers' comments:

Reviewer's Responses to Questions

**Comments to the Author**

1. Is the manuscript technically sound, and do the data support the conclusions?

Reviewer #1: Yes

Reviewer #2: Partly

2. Has the statistical analysis been performed appropriately and rigorously? 

Reviewer #1: Yes

Reviewer #2: Yes

3. Have the authors made all data underlying the findings in their manuscript fully available?

Reviewer #1: Yes

Reviewer #2: Yes

4. Is the manuscript presented in an intelligible fashion and written in standard English?

Reviewer #1: Yes

Reviewer #2: No

5. Review Comments to the Author

Reviewer #1: The study is relevant in assessing the complication readiness and birth preparedness of expectant mothers. The significance here is that arrangements can be put in place transport to hospital before complications set in. Also source of additional funds can be identified and blood donors put on standby. The study provides strategy for reducing maternal mortality.

Reviewer #2: 1. There are methodological issues that needs to be addressed by the Authors. Detail explanation can be found in the attached file

2. Yes, although further explanation has to be provided for the sake of clarity

3. Yes

4. No, requires serious proofreading and English proficiency service

6. PLOS authors have the option to publish the peer review history of their article (what does this mean?). If published, this will include your full peer review and any attached files.

Reviewer #1: No

Reviewer #2: No

---

## [Author Response · Author response to Decision Letter 0]

9 Mar 2022

Dear editors and reviewers, Thank you for your helpful and constructive comments. I have a great appreciation for ideas you raised during review time. Responses to the specific reviewer and editor comments were fully explained in the rebuttal letter, which labeled as 'Response to the Reviewer' and uploaded separately.

---

## [Decision Letter · Decision Letter 1]

24 Jun 2022

PONE-D-21-29865R1Knowledge of postpartum complications and associated factors among women gave birth in the last 12 months in Arba Minch Town, Southern Ethiopia, 2019: Community Based Cross-Sectional StudyPLOS ONE

Dear Dr. Yaya,

Thank you for submitting your manuscript to PLOS ONE. After careful consideration, we feel that it has merit but does not fully meet PLOS ONE’s publication criteria as it currently stands. Therefore, we invite you to submit a revised version of the manuscript that addresses the points raised during the review process.

ACADEMIC EDITOR: :

The Manuscript has undergone extensive revsion but still requires Enflish Language editing.Additonally,the reference section should be revised and ensure all references are written according to the Vancouver referencing style.Also ensure journames names are properly abbrevaited as they appear in the list of index medicus

.==============================

We look forward to receiving your revised manuscript.

Kind regards,

Godwin Otuodichinma Akaba, MBBS,MSc,MPH,FWACS

Academic Editor

PLOS ONE

Journal Requirements:

Additional Editor Comments:

The manuscript has undergone some extensive revisions but will still need English Language editing. Additionally, the references should be revised in conformance to the vancouver referencing style. The journal names should be listed as they appear in the list of index medicus.

Reviewers' comments:

Reviewer's Responses to Questions

**Comments to the Author**

1. If the authors have adequately addressed your comments raised in a previous round of review and you feel that this manuscript is now acceptable for publication, you may indicate that here to bypass the “Comments to the Author” section, enter your conflict of interest statement in the “Confidential to Editor” section, and submit your "Accept" recommendation.

Reviewer #2: All comments have been addressed

Reviewer #3: (No Response)

2. Is the manuscript technically sound, and do the data support the conclusions?

Reviewer #2: Yes

Reviewer #3: Yes

3. Has the statistical analysis been performed appropriately and rigorously? 

Reviewer #2: (No Response)

Reviewer #3: Yes

4. Have the authors made all data underlying the findings in their manuscript fully available?

Reviewer #2: Yes

Reviewer #3: No

5. Is the manuscript presented in an intelligible fashion and written in standard English?

Reviewer #2: No

Reviewer #3: No

6. Review Comments to the Author

Reviewer #2: Dear Editor,

Thank you for the opportunity to review this paper.

I still strongly recommend the review of this manuscript by a proofreading service or by a fluent English researcher. It still requires an extensive proofreading before publication.

Reviewer #3: 1. typographical errors present

2. the reference style is not uniform for all the references

3. Tables and charts not seen

7. PLOS authors have the option to publish the peer review history of their article (what does this mean?). If published, this will include your full peer review and any attached files.

Reviewer #2: No

Reviewer #3: No

---

## [Author Response · Author response to Decision Letter 1]

21 Jul 2022

Dear editors and reviewers, Thank you for your helpful and constructive comments. I

have a great appreciation for ideas you raised during review time. Responses to the reviewer and editor comments were fully explained in the rebuttal letter, which

labeled as 'Response to the Reviewer' and uploaded separately

---

## [Decision Letter · Decision Letter 2]

21 Sep 2022

PONE-D-21-29865R2Knowledge of postpartum complications and associated factors among women gave birth in the last 12 months in Arba Minch Town, Southern Ethiopia, 2019: A Community Based Cross-Sectional StudyPLOS ONE

Dear Dr. Yaya,

Thank you for submitting your manuscript to PLOS ONE. After careful consideration, we feel that it has merit but does not fully meet PLOS ONE’s publication criteria as it currently stands. Therefore, we invite you to submit a revised version of the manuscript that addresses the points raised during the review process. Thank you for revising your manuscript.I observed that there are still issues raised during the last revsiuons which you have not yeet addressed.Additonally,the journal names have not yet been properly as they appear in the list of index medicus.One of the reviwers have also highlighted the need to conpletely resolve the grammatical errors noted in the manuscript.You will need to properly adress these issues. 

We look forward to receiving your revised manuscript.

Kind regards,

Godwin Otuodichinma Akaba, MBBS,MSc,MPH,FWACS

Academic Editor

PLOS ONE

Journal Requirements:

Additional Editor Comments:

Abstract

Comment: Please delete: 2019: Community-Based cross-sectional Study

Data collection: and was translated……….

Data quality management: Before data collection, a semi-structured questionnaire was adapted from previously published kinds of literature.

Comment: Please correct to: Before data collection, a semi-structured questionnaire was adapted from previously published literature.

Comment: Also reference the literature accordingly

Results

. Breast complications were mentioned 75 (17.9 percent), blurred vision and weakness were mentioned 73 (17.5%), foul-smelling vaginal discharge was mentioned 71 (17%), convulsions were mentioned 14 (3.3%), and postpartum psychoses and blues were mentioned the least 4 (1%).

Comment: Paraphrase to “. Breast complications were mentioned in 75 (17.9 %), blurred vision and weakness; 73 (17.5%), foul-smelling vaginal discharge; 71 (17%), convulsions ;14 (3.3%), and postpartum psychoses and blues were mentioned the least in 4 (1%).

Discussion

Page 9: This difference might be mothers who could mention two and above complications were considered as knowledgeable.

Comment: Paraphrase to: This difference might be because mothers who could mention two and above complications were considered as knowledgeable in the previous study.

Page 10: This is similar to the institutional-based cross-sectional (2014) study finding of Mechekel East Gojjam Zone, Ethiopia [25].

Comment: Paraphrase to : This is similar to the institutional-based cross-sectional study finding of Mechekel East Gojjam Zone, Ethiopia [25].

Page 10, last line: This is in line with study findings conducted in Arba Minch town, Ethiopia; that test association for general knowledge of obstetrical complications of the three periods [24].

Comment: Change to: This is in line with study findings conducted in Arba Minch town, Ethiopia; that studied knowledge of obstetric danger signs and associated factors [24].

References:

The journal names in the quoted references have not been written properly as they appear in the list of index medicus as previously recommended in the last decision to authors

Reviewers' comments:

Reviewer's Responses to Questions

**Comments to the Author**

1. If the authors have adequately addressed your comments raised in a previous round of review and you feel that this manuscript is now acceptable for publication, you may indicate that here to bypass the “Comments to the Author” section, enter your conflict of interest statement in the “Confidential to Editor” section, and submit your "Accept" recommendation.

Reviewer #3: All comments have been addressed

2. Is the manuscript technically sound, and do the data support the conclusions?

Reviewer #3: Yes

3. Has the statistical analysis been performed appropriately and rigorously? 

Reviewer #3: Yes

4. Have the authors made all data underlying the findings in their manuscript fully available?

Reviewer #3: Yes

5. Is the manuscript presented in an intelligible fashion and written in standard English?

Reviewer #3: No

6. Review Comments to the Author

Reviewer #3: 1. There are still quite a few grammatical errors.

2. The methodology needs to be a little bit more explicit.

3. The results needs to be worked on further.

4. The discussion is not in depth.

7. PLOS authors have the option to publish the peer review history of their article (what does this mean?). If published, this will include your full peer review and any attached files.

Reviewer #3: No

---

## [Author Response · Author response to Decision Letter 2]

9 Nov 2022

All of your comments were constrictive, and which were built my article. More over, I have wrote my response in the response to the reviewers letter.

---

## [Decision Letter · Decision Letter 3]

19 Jan 2023

Knowledge of postpartum complications and associated factors among women gave birth in the last 12 months in Arba Minch Town, Southern Ethiopia, 2019: A Community Based Cross-Sectional Study

PONE-D-21-29865R3

Dear Dr. Yaya,

We’re pleased to inform you that your manuscript has been judged scientifically suitable for publication and will be formally accepted for publication once it meets all outstanding technical requirements.

Kind regards,

Tesera Bitew, PhD

Academic Editor

PLOS ONE

Additional Editor Comments (optional):

Reviewers' comments:

Reviewer's Responses to Questions

**Comments to the Author**

1. If the authors have adequately addressed your comments raised in a previous round of review and you feel that this manuscript is now acceptable for publication, you may indicate that here to bypass the “Comments to the Author” section, enter your conflict of interest statement in the “Confidential to Editor” section, and submit your "Accept" recommendation.

Reviewer #3: All comments have been addressed

2. Is the manuscript technically sound, and do the data support the conclusions?

Reviewer #3: Yes

3. Has the statistical analysis been performed appropriately and rigorously? 

Reviewer #3: Yes

4. Have the authors made all data underlying the findings in their manuscript fully available?

Reviewer #3: Yes

5. Is the manuscript presented in an intelligible fashion and written in standard English?

Reviewer #3: Yes

6. Review Comments to the Author

Reviewer #3: The writers have made the necessary corrections. The typographical and grammatical errors have been corrected. The statistical analysis has also been done well.

7. PLOS authors have the option to publish the peer review history of their article (what does this mean?). If published, this will include your full peer review and any attached files.

Reviewer #3: No
